

# Weekly water quality monitoring data for the River Thames (UK) and its major tributaries (2009-2013): The Thames Initiative research platform

Michael J. Bowes[1, *], Linda K. Armstrong[1], Sarah A. Harman[1], Heather D. Wickham[1], Peter M. Scarlett[1], Colin Roberts[1], Helen P. Jarvie[1] Gareth Old[1], Emma Gozzard[1], Nuria Bachiller-Jareno, Daniel S. Read[1]

[1]Centre for Ecology & Hydrology, Benson Lane, Crowmarsh Gifford, Wallingford, Oxfordshire, OX10 8BB, UK

*Correspondence to*: Michael J. Bowes (mibo@ceh.ac.uk)

## Abstract

The River Thames and 15 of its major tributaries have been monitored at weekly intervals since March 2009. Monitored determinands include major nutrient fractions, anions, cations, metals, pH, alkalinity and chlorophyll *a*., and linked to mean daily river flows at each site. This catchment-wide biogeochemical monitoring platform captures changes in the water quality of the Thames basin during a period of rapid change, related to increasing pressures (due to a rapidly growing human population, increasing water demand and climate change) and improvements in sewage treatment processes and agricultural practises. The platform provides the research community with a valuable data and modelling resource for furthering our understanding of pollution sources and dynamics, and interactions between water quality and aquatic ecology. Comparing Thames Initiative data with previous (non-continuous) monitoring data sets from many common study sites, dating back to 1997, has shown that there have been major reductions is phosphorus concentrations at most sites, occurring at low river flow, and these are principally due to reduced loadings from sewage treatment works. This ongoing monitoring programme will provide the vital underpinning environmental data required to best manage this vital drinking water resource, which is key for the sustainability of the city of London and the wider UK economy. The Thames Initiative data set is freely available from the Centre for Ecology & Hydrology's Environmental Information Data Centre at doi:10.5285/e4c300b1-8bc3-4df2-b23a-e72e67eef2fd.

## 1 Introduction

The River Thames and its tributaries play a vital role in sustaining the ca. 13 million inhabitants of the Thames basin, including the UK capital, London. The river system supplies most of their drinking water, and also provides a means of exporting human and industrial wastes, by receiving and transporting wastewater treatment effluents. The rivers also provide important recreational services, being extensively used for fishing, walking and boating. The river network therefore plays an important underpinning role to support the UK economy. However, the River Thames basin faces growing pressures from a rapidly increasing population and water usage, which is likely to increase pollution loadings and water stress in future decades. These





effects are likely to be exacerbated by future climate change, with predicted lower flows and droughts in the Thames region in the summer months and increased flooding in the winter (Bell et al., 2012; Johnson et al., 2009). To reduce these effects, major mitigation measures are being implemented. Diffuse pollution from agricultural activities is being targeted through schemes such as the Catchment Sensitive Farming Initiative, and point source inputs have been reduced through large-scale

investment in improved levels of sewage treatment (Kinniburgh et al., 1997), to comply with the Urban Waste Water Treatment Directive (EEC, 1991) and Water Framework Directive (CEC, 2000).

The River Thames is one of the UK's most monitored and studied rivers. Due to its importance as a source of drinking water for London, the lower River Thames has been continuously monitored for nitrate concentration stretching back to 1868;

perhaps the longest continuous water quality record in the world (Howden et al., 2010). The corresponding phosphorus record for the River Thames goes back to 1936 (Haygarth et al., 2014; Powers et al., 2016). Environment Agency regulatory monitoring of phosphorus concentrations in the River Thames since the 1970s has been used to identify the major improvements in water quality due in part to the introduction of the Urban Waste Water Treatment Directive (Kinniburgh et al., 1997).

Phosphorus and nitrogen sources and dynamics in the River Thames (Neal et al., 2010a; Neal et al., 2000c) and its tributaries (Bowes et al., 2012c; Jarvie et al., 2006; Jarvie et al., 2002b; Neal et al., 2000a; Neal et al., 2004; Neal et al., 2006), and how nutrients interact with river ecology (House et al., 2001; Jarvie et al., 2002a; Williams et al., 2000) have been intensively studied in recent decades. These studies were based on one to two year periods of weekly monitoring by the Centre for Ecology & Hydrology (CEH), based at a small number of individual study sites, carried out between 1997 and 2008. These data are

freely         available         from         the         CEH         Environmental         Information         Data         Centre         portal         at https://catalogue.ceh.ac.uk/documents/8e23a86b-6b54-4564-9789-23f4b4e045ea (Neal et al., 2012). Extensive biological surveys of phytoplankton (Lack, 1971; Lack and Berrie, 1975), macrophytes (Flynn et al., 2002), macroinvertebrates (Wright et al., 2002), zooplankton (May and Bass, 1998) and fish stocks (Mann et al., 1972) have also been conducted across the catchment since the 1960s.


The above studies were based on monitoring of a limited numbers of sites for one- to two-year durations. Therefore, step changes in water quality were often missed, and it was not possible to determine if observed changes in water quality were due to basin-wide conditions (such as weather conditions) or a specific change in catchment land use / management. Therefore, there was a strategic need to conduct continuous water quality monitoring at multiple sites across the Thames catchment,

covering a large range of water quality determinands. The resulting monitoring platform was the CEH Thames Initiative. A weekly temporal resolution was adopted, as this was deemed appropriate to capture algal blooms and a selection of high flow events (Bowes et al., 2009). This was supplemented by hourly phosphorus autoanalyser and nitrate probe data (alongside a range of other water quality data) at individual river sites (Bowes et al., 2015; Halliday et al., 2015) (Data freely available from the CEH Environmental Information Data Centre portal at https://catalogue.ceh.ac.uk/documents/db695881-eabe-416c-b128-



76691b2104d8). The spatial resolution of the monitoring platform needed to cover the impacts of a range of river sizes, land use types and management options. Therefore the Thames Initiative consists of sites along the length of the freshwater Thames, plus its major tributary sub-catchments. The third feature of the Thames Initiative was that it would characterise aquatic ecology (particularly phytoplankton and bacterioplankton communities) at the same weekly frequency as the water chemistry.

The fourth feature of the Thames Initiative has that it needed to be long term (decadal), to identify water quality impacts in response to land use change, population pressure, sewage treatment works (STW) improvements, and variations in climate.

Since its commencement in March 2009, the Thames Initiative data has been used to quantify nutrient dynamics (Bowes et al., 2015; Skeffington et al., 2015; Wade et al., 2012), nutrient sources (Bowes et al., 2014), phytoplankton dynamics (Bowes et

al., 2012a; Bowes et al., 2016; Read et al., 2014) and bacterioplankton biodiversity (Read et al., 2015). The resulting data sets have been extensively used as a modelling resource (Bussi et al., 2017; Bussi et al., 2016; Hutchins et al., 2016; Whitehead et al., 2015). Other studies have used the Thames Initiative monitoring sites as a framework for investigations of organic pollutants (Nakada et al., 2017; Singer et al., 2014), heavy metals (Turner et al., 2014), emerging contaminants (Horton et al., 2017), and antimicrobial resistance (Amos et al., 2015).

The objective of this paper is to present an overview of the comprehensive data produced by the Thames Initiative research platform, detailing how samples were taken and analysed, and providing a general description of the spatial and temporal patterns in water quality. We also provide some new basic interpretation of the data, to enable future users of the data to place their studies within this basic framework.

**2 Sampling and analytical methodology**

**2.1 Design of monitoring programme**

The River Thames was monitored at weekly intervals at seven points along its length, extending from the upper Thames in the Cotswold Hills (at Hannington Wick, 46 km from the source) to Runnymede (23 km upstream of the tidal limit), covering a monitored reach of 189 km. In the lower reach, just upstream of the town of Slough, some of the water of the River Thames is

channelled along the Jubilee River, which is an artificial, engineered channel that is used for flood defence for the towns of Maidenhead and Windsor. One of the seven River Thames monitoring sites is on this Jubilee River, just before it re-joins the River Thames. In addition, 15 of the major tributaries entering this monitored stretch of the River Thames were also monitored. Sites were selected to be easily accessible by road (at road bridges wherever possible, to allow sampling from the middle of the river) and close to their confluences with the River Thames. The locations of the sampling sites are given in Table 1 and

Figure 1. Most monitoring sites were located at, or very near to Environment Agency flow gauging stations. The monitoring programme began at most sites at the beginning of March 2009, with monitoring of the River Thames at Hannington Wick





commencing in October 2009, the Rivers Kennet and Enborne in November 2009, and the Jubilee River in January 2010. Data is presented up to the end of February 2013, although the monitoring programme is ongoing.

## 2.2 River sampling

Bulk samples were taken from the main flow of each river, using a plastic bucket on a rope. The bucket was rinsed twice with the local river water prior to sampling, to avoid cross-contamination between sites. Sampling of the sites took place on either Monday or Tuesday of each week. The bulk samples were immediately subsampled into two 500 ml bottles for suspended solids and chlorophyll analysis, an amber-coloured glass bottle (filled to the brim and sealed to minimise degassing) for pH and alkalinity determinations, and into 60 ml bottles for total metals and total phosphorus analysis. Other sub-samples were
filtered immediately in the field through a 0.45 μm cellulose nitrate (Whatman WCN grade; Maidstone, UK) membrane filter into 60 ml bottles, for dissolved metals, nutrients, anions and cations analysis. All bottles were acid-washed prior to use. The water temperature of the bulk river water sample was measured in the field using an ATP Multi-Thermo digital thermometer (ATP Instrumentation Ltd. Ashby-de-la-Zouch, UK).

## 2.3 Analytical methods

On return to the laboratory, all samples were stored in the dark at 4ºC, prior to analysis. The pH was determined using a Radiometer Analytical PHM210 pH meter. The instrument was calibrated prior to use using pH 4, 7, and 10 buffer solutions traceable to National Institute of Standards and Technology (Gaithersburg, USA). Gran alkalinity was determined by acidimetric titration to pH 4 and 3 using 0.5N $H_2SO_4$. Suspended solids concentrations were determined by filtering a known volume (approximately 500 ml) of river water through a pre-dried Whatman GF/C filter paper. The filter paper was then re-
dried (16 h at 80ºC) and reweighed to determine the mass of solids in the water sample. Chlorophyll concentrations were determined by filtering a known volume of unfiltered river water (approximately 500 ml) through a Whatman GF/C filter paper. The filter paper was then extracted in 10 ml of 90% v/v acetone/water and refrigerated overnight at 4ºC in the dark. Chlorophyll-a concentration was determined colorimetrically using a Beckman 750 DU spectrophotometer, using the method of Marker et al. (1980). Chlorophyll analysis was completed within 48 hours of sampling, to avoid errors due to sample
stability. Total phosphorus (TP) and total dissolved phosphorus (TDP) were determined by digesting an unfiltered and 0.45 μm filtered water sample (respectively) with acidified potassium persulphate in an autoclave at 121°C for 45 min. Acidified ammonium molybdate reagent was then added to the digested samples to produce a molybdenum–phosphorus complex. This intensely blue-coloured compound was then quantified spectrophotometrically at 880 nm (Eisenreich et al., 1975). Soluble reactive phosphorus (SRP) concentrations were determined on a filtered (0.45 μm WCN-grade cellulose nitrate membrane;
Whatman, Maidstone, UK) sample, using the phosphomolybdenum-blue colorimetry method of Murphy and Riley (1962), as modified by Neal et al. (2000b), using a Seal Auto Analyser 3 (Seal Analytical; Fareham, UK). SRP samples were analysed within 48 h, to minimise errors associated with sample instability. Prior to August 16[th] 2010, dissolved reactive silicon





concentrations were determined by reaction with acid ammonium molybdate, to form yellow molybdosilicic acids. These were then reduced using an acidified tin (II) chloride solution to form intensely coloured silicomolybdenum blues, which were quantified spectrophotometrically using a Seal Auto Analyser 2 (Seal Analytical; Fareham, UK) (Mullin and Riley, 1955). From 16th August 2010 onwards, dissolved reactive silicon was determined by addition of acid ammonium molybdate, to again

form yellow-coloured molybdosilicic acids, and then oxalic acid was added (to eliminate phosphate interference) followed by ascorbic acid to reduce the yellow compound to molybdenum blue, which was quantified using a Seal Auto Analyser 3 spectrophotometer (Skougstad et al., 1978).  Ammonium concentration was determined using an indophenol-blue colorimetric method (Leeks et al., 1997) using a Seal Auto Analyser 3. Dissolved organic carbon and total dissolved nitrogen were analysed by thermal oxidation using a Thermalox analyser (Analytical Sciences Ltd.; Cambridge, UK) until December 2010 and with

an Elementar Vario Cube (Elementar Ananlysensysteme GmbH; Langenselbold, Germany) from June 2011.  There are large gaps in the DOC data set between May 2010 and May 2011, due to instrument breakdowns. Major dissolved anion (fluoride, chloride, bromide, nitrite, nitrate and sulphate) concentrations were determined by ion chromatography (Dionex AS50, Thermo Fisher Scientific; Waltham, USA). Total and dissolved cation concentrations were determined on unfiltered and filtered samples respectively, by acidification, followed by analysis by inductively coupled plasma optical emission spectrometry

(ICP-OES)(Perkin Elmer Optima 2100; Seer Green, UK).  All analyses (with the exception of suspended solids) were carried out alongside reference Aquacheck quality control standards (LGC Standards, Teddington, UK).

## 2.4 River flow data

The water quality data sets are presented alongside the mean daily river flows for the sites on the day of sampling. The mean daily flow data was downloaded from the National River Flow Archive (http://nrfa.ceh.ac.uk/) in January 2017.  Most sites

were co-located at, or very close to, Environment Agency flow gauging stations. The exceptions were the River Kennet (flow gauging data taken from Theale, approximately 5 km downstream of the water quality sampling site at Woolhampton) and The Cut (flow gauging data taken from Binfield, which is approximately 6 km upstream of the water quality sampling site of Paley Street).  The other exceptions were along the River Thames itself. The Thames water quality monitoring sites at Hannington Wick, Newbridge, Wallingford and Sonning had their mean daily flows estimated by interpolation of adjacent River Thames

flow gauging sites data, adjusted based on the catchment area of the water quality sampling site (Bowes et al., 2014).

## 2.5 Site characterisation

The characteristics of each sub-catchment, upstream of each sampling point, were determined using GIS (Table 1).  The physical characteristics of the catchments were determined using the Flood Estimation Handbook web service (https://fehweb.ceh.ac.uk/).  Land use percentage cover and total upstream STW population estimates (PE) were determined

in ARC GIS using the CEH Intelligent River Network (Dawson et al., 2002) and UK Land Cover Map 2000 (Fuller et al.,



2002), using the RACQUEL web application. Base flow index data were derived from NRFA flow data, obtained from the UK Hydrometric Register (Marsh and Hannaford, 2008).

The monitoring platform encompasses a wide range of river sizes, from short streams of less than 40 km in length with mean flows of less than 1 $m^3 s^{-1}$ (River Wye, Leach and Pang) to the lower reaches of the River Thames, with mean annual flows of

*ca.* 58 $m^3 s^{-1}$ (Marsh and Hannaford, 2008) and a river length of 240 km at Runnymede. The monitoring programme also covers an extremely wide range of population pressures, with the Coln and Leach catchments in the Cotswolds region being very rural, with less than 3% urban and semi-urban land cover (Fuller et al., 2002). In contrast, other tributaries (The Cut and River Lodden and Wye) have >20% urban and semi-urban land cover, and very high population densities connected to the wastewater treatment infrastructure (>500 STW PE $km^{-2}$). The River Thames itself is impacted by sewage for the entire

monitored stretch, from the large towns and cities of Swindon, Oxford, Reading and Slough, alongside hundreds of small towns and village STWs all discharging wastewater along its length.

## 3 Results and discussion

### 3.1 General water quality characteristics

Due to the chalk and limestone geology underlying most of the catchment, the River Thames and most of its tributaries have very high Gran alkalinities (in excess of 3000 µEq $l^{-1}$) and high pH (mean values ranging from of 7.66 to 8.07) (Table 2). Most of the river flows comprise of substantial groundwater inputs, with base flow index values ranging from 0.95 (River Coln) to 0.60 (River Thame) (Table 1). Some of the catchments (such as the Rivers Enborne, Ray and Cole) are overlaid with clay

deposits, and these rivers have lower base flow indexes of between 0.53 and 0.57, due to increased quantities of run-off. The Cut is an artificial river that was diverted from the upper River Lodden, and has the lowest base flow index (0.46), and also the lowest mean Gran alkalinity of 2450 µEq $l^{-1}$. Mean suspended solids concentrations across the basin were relatively low (ranging from 3.0 to 15.5 mg $l^{-1}$) (Table 2), due to the low catchment gradients and groundwater dominance of the flow regime. The lowest suspended solids concentrations were observed in the small tributaries of the Cotswolds (Rivers Coln and Leach).

The highest suspended solids concentrations were found in some of the clay-covered catchments (Rivers Thame and Cole) and the steepest catchment; the River Wye. Monitoring points along the length of the River Thames and the River Cherwell also had relatively high suspended solids concentration, but much of this would be due to high phytoplankton biomass through the spring and summer, rather than sediment entrainment / soil erosion.

### 3.2 Spatial data

All monitoring sites showed significant nutrient enrichment, which reflects the high population densities across much of the catchment, the large number of STWs discharging into the River Thames and its tributaries, and relatively intensive agricultural



activity in the region. Eighteen of the 22 monitoring sites had average SRP concentrations greater than 100 µg P $l^{-1}$, with the Rivers Ray, Thame and Cut in excess of 400 µg P $l^{-1}$ (Table 3, Figure 2). The monitoring sites along the River Thames had relatively consistent levels of phosphorus enrichment (with mean concentrations varying from 116 to 212 µg SRP $l^{-1}$ and 171 to 301 µg TP $l^{-1}$), due to it receiving significant STW inputs from major towns along its entire length. River Thames average

phosphorus concentrations were highest in the middle reach (at Wallingford), due to its position downstream of both Oxford STW (Figure 1) and the inputs from the phosphorus-polluted tributaries of the River Thame and Ray (Figure 2). Only three catchments had low SRP concentrations which may be potentially limiting for primary production (Bowes et al., 2012b; Bowes et al., 2010; McCall et al., 2017); the Rivers Leach, Kennet and Pang, which had average SRP concentrations of 20, 34 and 38 µg P $l^{-1}$ respectively (Table 3, Figure 2). This is probably due to the Rivers Pang and Leach having the lowest STW population

equivalent densities (30 and 20 PE $km^{-2}$ respectively), and although the River Kennet has a higher STW-PE density (114 PE $km^{-2}$), all STWs along its length have had tertiary phosphorus stripping installed, with most final effluent consents set at <1000 µg P $l^{-1}$. However, these three rivers are still highly nutrient impacted, due to their very high nitrate concentrations of between 24 and 31 mg $NO_3$ $l^{-1}$.

All sites were heavily polluted with nitrate, with 20 of the 22 sites having nitrate concentrations in excess of 20 mg $NO_3$ $l^{-1}$

(Table 3, Figure 2). This is mainly due to historic agricultural contamination of the groundwaters which commonly occurs in chalk-dominated catchments (Smith et al., 2010). Due to high catchment porosity, manures and fertilisers are easily transported from the land surface and into the groundwater aquifers. Nitrate concentrations in the lower Thames have increased from ca. 8 mg $NO_3$ $l^{-1}$ in the 1880s to 20 mg $NO_3$ $l^{-1}$ by the 1950s (attributed to ploughing up of grassland in the 1940s, which resulted in large-scale mineralisation of organic N), and increased rapidly to ca. 34 mg $NO_3$ $l^{-1}$, due to increased fertiliser applications

and increasing arable land cover in the 1960s (Howden et al., 2010). By far the highest average nitrate concentration was observed in The Cut, which, at 83.7 mg $NO_3$ $l^{-1}$, was over twice as high as any of the other sites. This gross nitrate pollution is most likely due to the high sewage loading at this site (with the highest STW population density of 1644 PE $km^{-2}$), as The Cut has the lowest base flow index and therefore the lowest groundwater inputs. The two other sites with the least groundwater inputs (BFI values less than 0.54; the Rivers Cole and Enborne) had the lowest average nitrate concentrations of 18.5 and 17.1

mg $NO_3$ $l^{-1}$ respectively (Figure 2). The average ammonium concentrations also reflect sewage inputs, with the five sites with average concentrations ≥1.0 mg $NH_4^+$ $l^{-1}$ (River Thames at Hannington Wick, the Rivers Wye, Ray, Thame, and the Cut) all having high STW-PE densities, and many were just downstream of major sewage works.

The average DOC concentrations varied from ≤ 3 mg C $l^{-1}$ for the relatively rural rivers in the Cotswolds (Rivers Leach, Coln and Evenlode) and the area west of the town of Reading (Rivers Pang and Kennet), to ≥ 8 mg C $l^{-1}$ for the sewage-impacted

River Ray and Cut. The clay-dominated sub-catchments of the River Enborne and Thame also had relatively high DOC concentrations (> 6 mg C $l^{-1}$), and may reflect the higher rates of runoff that possibly wash greater quantities of organic material from the catchment into the river, compared to the other groundwater-dominated monitoring sites. The spatial pattern in the average dissolved boron concentrations also reflected the pattern in sewage inputs, with some of the most sewage-impacted tributaries (Rivers Thame, Ray, Cherwell and The Cut) having concentrations in excess of 70 µg B $l^{-1}$ (Table 4, Figure 2). This



is because boron is a constituent of detergents, and has been used as a sewage tracer in river research in the past, although concentrations are declining rapidly due to changing detergent formulations (Neal et al., 2010b). The highest average boron concentration in the River Thames was observed in the middle reaches at Wallingford, due to its location downstream of Oxford STW and the confluences with the Rivers Ray, Cherwell and Thame, all with significant boron loads. The lowest concentrations were again observed in the rural tributaries of the Cotswolds (Rivers Coln and Leach), and the Rivers Pang, Enborne and Kennet, west of Reading. The highest manganese, zinc and copper concentrations were observed in The Cut (Table 4), due to it having by far the highest STW population density (Table1). Dissolved sodium concentration has a strong correlation with STW-PE density (Pearson Correlation Coefficient = 0.785), indicating the element's suitability as a sewage tracer. Its spatial pattern is similar to boron concentration (Figure 2), with highest concentrations in The Cut, Thame, Cherwell and the upper Thames downstream of Swindon STW, and lowest in the most rural sites.

The spatial pattern in average chlorophyll concentrations was in sharp contrast to the nutrient concentrations and sewage indicators described above. Highest concentrations were observed along the middle and lower reaches on the River Thames, and in the larger tributaries such as the Cherwell, Evenlode and Thame. There is a strong positive relationship with the distance from the monitoring point to the river source, which is probably linked with the long transit times required to develop substantial phytoplankton biomass. The spatial and temporal patterns in these chlorophyll concentration data across the Thames catchment is discussed at length in (Bowes et al., 2012a).

### 3.3 Temporal data

The time series plots for the lower River Thames at Runnymede are presented as an example in Figures 3 to 5, demonstrating the continuous nature of the data. The data sets from all other sites are equally complete.

### 3.3.1 Physical data

River water temperature follows a relatively consistent pattern of highest temperatures in the August of each year, and lowest temperatures occur between December and February (Figure 3). The pH data varies between 7.6 and 8.2, reflecting the alkaline bedrock throughout the catchment. It is important to note that previous high-frequency monitoring of rivers across the Thames catchment have shown marked diurnal pH fluctuations of up to 0.6 pH units (Halliday et al., 2014; Halliday et al., 2015), and therefore caution should be exercised when using data from weekly manual samples. The Gran alkalinity was high (mean = 3999 µeq l$^{-1}$), due to the chalk and limestone bedrock. Short-term reductions in Ca and alkalinity (to below 70 mg Ca l$^{-1}$ and 2600 µeq l$^{-1}$) in the Thames at Runnymede coincided with peaks in chlorophyll concentrations in 2009, 2011 and 2012 (Figure 3). During algal blooms high rates of photosynthesis result in depletion of dissolved carbon dioxide and calcium carbonate precipitation can occur (Hartley et al., 1995; Neal et al., 2002), resulting in reductions in alkalinity and Ca concentration within the water column. Suspended solid concentrations were highest during periods of high flow, due to a combination of soil erosion inputs during the wet winter and spring periods, and the resuspension of bed sediment within the river channel. Suspended solid concentration peaks in the lower River Thames also coincided with peaks in chlorophyll concentration,



indicating that phytoplankton biomass provided a significant proportion of the suspended solids load during spring / summer algal blooms. Peaks in chlorophyll concentrations occurred between April and June of each year, but the magnitude and duration of chlorophyll peaks varied greatly between years. The timing and magnitude of blooms has been shown to be related to SRP and dissolved silicon concentrations, sunlight duration, river flow and water temperature, and is fully described in

Bowes et al. (2016). The largest and sustained chlorophyll peaks are observed in the middle and lower Thames sites, and the longer tributaries (River Cherwell, River Thame), especially those connected to canal systems, indicating the importance of residence time (Bowes et al., 2012a). The river flow data shows a regular pattern, typical of UK rivers, of highest flows over the winter / spring periods (December to February), and lowest flows in July to October. However, the data set also captures a winter drought in 2011/2012, followed by uncharacteristic flooding in summer 2012.

**3.3.2 Nutrient data**

Phosphorus in the River Thames at Runnymede was predominantly in SRP form (Figure 4), indicating the dominance of sewage effluent inputs to the River Thames and many of its tributaries. However, during periods of high chlorophyll concentrations in 2009, 2011 and 2013, SRP concentrations reduced to below 20 µg P $l^{-1}$ while TP was maintained at ca. 200 µg $l^{-1}$, indicating that dissolved phosphorus was being sequestered by the rapidly growing phytoplankton biomass (thereby

becoming particulate phosphorus). The two highest peaks in TP concentration (30th April and 26th November 2012; Figure 4) correspond with the two highest suspended solids concentrations (Figure 3), occurring during the major storm that ended the 2011/2012 winter drought and the highest recorded river flow respectively. This indicates that there were major inputs of particulate-bound phosphorus from the catchment during these storm events, particularly following dry antecedent conditions. Nitrate concentrations remained high throughout the monitoring period, as a result of the gross pollution of groundwater

aquifers typical of English chalk catchments (Smith et al., 2010). Small decreases in nitrate concentration coincided with flow peaks, indicating that these high nitrate inputs from groundwater were being diluted by rainwater / surface runoff inputs to the River Thames. Other more-sustained declines in nitrate concentrations coincided with periods of high chlorophyll concentrations and SRP depletion in spring of 2009 and 2011, due to N uptake by phytoplankton. Dissolved reactive silicon also depleted to below 1 mg $l^{-1}$ during these phytoplankton blooms, indicating that diatoms (with their silicon frustules) were

the major component of algal biomass. Nitrite concentrations were consistently low (<0.05 mg $NO_2^-$ $l^{-1}$), but were high during the winter drought period of 2011, possibly due to lack of dilution of sewage effluent inputs, and low biological processing rates on the nitrite within the river channel at low water temperatures. The highest ammonium concentrations (up to 0.26 mg $NH_4^+$ $l^{-1}$) was observed during the flooding in April 2012, due to in-wash of fertilisers and manures from agricultural fields into the watercourses.






### 3.3.3 Cation data

Sodium, potassium, calcium, magnesium and boron loads in the River Thames at Runnymede were present almost entirely in dissolved form (Figure 5). In contrast, iron was largely present in particulate form. Similar patterns were observed at all monitoring sites within this study. Sodium, potassium and to a lesser extent, boron, all showed sudden drops in concentration (Figure 5) coinciding with periods of high flow (Figure 3), indicating that their predominant sources (sewage effluent and groundwater) were being diluted by runoff and rainwater inputs. Particulate iron concentrations peaked during high flows, indicating that the predominant source was diffuse inputs from catchment soil erosion and resuspension of within-channel bed sediments. The calcium concentration data was relatively constant through the monitoring period, at ca. 100 mg Ca $l^{-1}$, but there were sudden reductions in calcium concentrations to below 80 mg $l^{-1}$ in the spring of each year, coinciding with reductions in alkalinity and periods of high chlorophyll concentrations, due to the precipitation of $CaCO_3$, as described above (3.3.1).

### 3.4 Long term temporal changes in nutrient concentrations

#### 3.4.1. Phosphorus

There have been no marked changes in water quality at any of the study sites over the 2009 to 2013 monitoring period of the Thames Initiative, indicating the lack of investment in sewage treatment improvements across the catchment through this period. However, comparisons with past monitoring data from the same study sites (Neal et al., 2012) (https://catalogue.ceh.ac.uk/id/8e23a86b-6b54-4564-9789-23f4b4e045ea) have shown that there has been a major reduction in both total phosphorus and soluble reactive phosphorus concentrations since the late 1990s at most sites.

An example for the middle reach of the River Thames at Wallingford is presented in Figure 6. Data prior to 2009 was collected from the River Thames at Howbery Park, which is approximately 200 m upstream of the River Thames at Wallingford monitoring site used during the Thames Initiative research platform. There are no known inputs between these monitoring sites, and so the data from each data set were equivalent. There is a major step reduction in SRP in 1998, and another reduction between 2002 and 2006 (Figure 6a), although the exact timing of this is unknown, as the monitoring data prior to the start of the Thames Initiative is not continuous. The Thames basin has been the focus for many mitigation measures aimed at reducing sewage effluent inputs of P (primarily through the Urban Wastewater Treatment Directive), and reducing diffuse P pollution from farming (e.g. through the Catchment Sensitive Farming Initiative and adoption of agri-environment schemes). The sudden reduction in SRP concentration in winter of 1998 / 1999 (halving the maximum annual SRP concentration from ca. 1800 µg SRP $l^{-1}$ to 800 SRP µg $l^{-1}$) strongly suggests that there were major interventions at this time, rather than the multiple small agricultural interventions. The relationship between SRP concentration and flow (Figure 6b) clearly shows that the reductions in SRP concentrations in both winter 1998 and prior to 2006 occurred during low flow periods. Again, this strongly suggests





that it is a constant P input (i.e. sewage effluent) that has been reduced, rather than rain-related agricultural inputs (Bowes et al., 2008). This is further confirmed by plotting the SRP concentrations against sodium concentration (a conservative sewage marker) (Neal et al., 2010b) (Figure 6b), which shows that the 80% reduction in SRP (from an average of 1016 µg l$^{-1}$ in 1997-1998 to an average of 212 µg l$^{-1}$ in 2009-2013) is predominantly due to reductions in sewage effluent phosphorus loadings.

This is further backed up by information on the implementation of the Urban Wastewater Treatment Directive within the catchment, which introduced phosphorus stripping at all STWs greater than 10,000 population equivalent between 1996 and March 2008).

However, these data sets also show that there has been little improvement in phosphorus concentrations in the River Thames

and its tributaries since February 2009. A previous study of the Thames Initiative data using Load Apportionment Modelling (Bowes et al., 2014) has highlighted that despite past improvements (due largely to the introduction of phosphorus stripping at sewage treatment works), the Thames and most tributaries are still dominated by STW P inputs. The most effective strategy to further reduce P concentrations across the catchment would be to focus resources at further reductions in point source inputs from STWs, rather than diffuse agricultural mitigation, especially as these data sets clearly show that STW improvements have

a major and immediate impact on river water quality.

### 3.4.2. Nitrate

The 140 year nitrate concentration data record of the lower River Thames by Howden et al. (2010) showed that nitrate concentrations have increased from <2 mg NO$_3$-N L$^{-1}$ in the late 1860s and 1870s, and reached a maximum of >7 mg NO$_3$-N L$^{-1}$ in the late 1970s to 1990s.  Since 2000, the average nitrate concentrations in the lower River Thames decreased slightly to

ca. 6.8 mg NO$_3$-N L$^{-1}$, suggesting that nitrate concentrations were beginning to decline.  The Thames Initiative data further supports this observation, showing that the average nitrate concentration in the lower River Thames at Runnymede (2009 – 2013) was 28.1 mg NO$_3$ L (Table 3), which is equivalent to 6.35 mg N / L.

Combining the CEH data sets from 1997 to 2008 and the CEH Thames Initiative data (2009 to 2013) provides good evidence

that nitrate concentrations have reached a turning point, and are beginning to slowly decline.  The middle River Thames at Wallingford shows nitrate concentrations peaking in 1998, and gradually reducing throughout the following years (Figure 7a). Nitrate concentrations showed little change over time at low flows (<25 m$^3$ s$^{-1}$) but were declining at medium to high flows (25 to 150 m$^3$ s$^{-1}$) (Figure 7b), suggestion that there is a reduction in diffuse, rain-related nitrate sources, but not constant inputs such as STWs.  This is further supported by examining the relationship between nitrate and sodium concentration. At high

sodium concentration > 40 mg l$^{-1}$ (indicating high sewage inputs) nitrate concentrations from the different monitoring periods are all relatively similar (Figure 7c). When sodium concentrations are lower (particularly ca. 20 mg l$^{-1}$), there has been a clear reduction in nitrate concentration throughout the monitoring periods.  These observations suggest that the reduction in river



nitrate concentration is due to reduced inputs from diffuse, rain-related sources such as agriculture run-off or groundwater nitrate concentration, and not due to reduced loadings from sewage effluent.

## 4. Conclusions

This catchment-wide biogeochemical monitoring platform provides the research community with a valuable data resource for furthering our understanding of pollution sources and dynamics, biological interactions, impacts of land use and increasing population pressures across this internationally-known river catchment.  The CEH Thames Initiative data can be linked with previous (non-continuous) monitoring data sets from many common study sites, dating back to 1997, and hourly physical and chemical data sets from two of its tributaries.  This ongoing monitoring programme will continue to capture the impacts of

increasing population densities, changes in agricultural practises and the impacts of improved sewage treatment processes, which have important implications for the sustainability of London and the UK economy.

## 5. Data availability

The entire data set presented in this study is freely available through the Centre for Ecology & Hydrology's Environmental Information Data Centre data portal https://catalogue.ceh.ac.uk/eidc/documents. The data set is titled "Weekly water quality

data from the River Thames and its major tributaries (2009-2013) [CEH Thames Initiative]".  The digital Object Identifier is doi:10.5285/e4c300b1-8bc3-4df2-b23a-e72e67eef2fd.

**List of Figures and Tables**

**Figure 1. Location of monitoring locations across the River Thames basin.**

**Figure 2. Spatial variation in chemical concentrations and chlorophyll-a across the Thames basin**

**Figure 3. General water quality time series data for the downstream extent of the monitoring platform (River Thames at Runnymede)**

**Figure 4. Nutrient concentration time series data for the downstream extent of the monitoring platform (River Thames at Runnymede)**

**Figure 5.  Cation concentration time series data for the downstream extent of the monitoring platform (River Thames at Runnymede)**

**Figure 6. Changes in soluble reactive P concentration in the middle reaches of the River Thames at Wallingford from 1997 to 2013, as (a) a time series, (b) related to mean daily river flow and (c) related to sodium concentration.**

**Figure 7. Changes in nitrate concentration in the middle reaches of the River Thames at Wallingford from 1997 to 2013, as (a) a time series, (b) related to mean daily river flow and (c) related to sodium concentration.**

**Table 1. Sampling locations, sampling periods and land-use statistics for catchments upstream of sampling sites.**



**Table 2. Mean water quality data from March 2009 to February 2013.**

**Table 3. Mean nutrient and chlorophyll-a concentrations from March 2009 to February 2013.**

**Table 4. Mean cation concentrations from March 2009 to February 2013.**

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



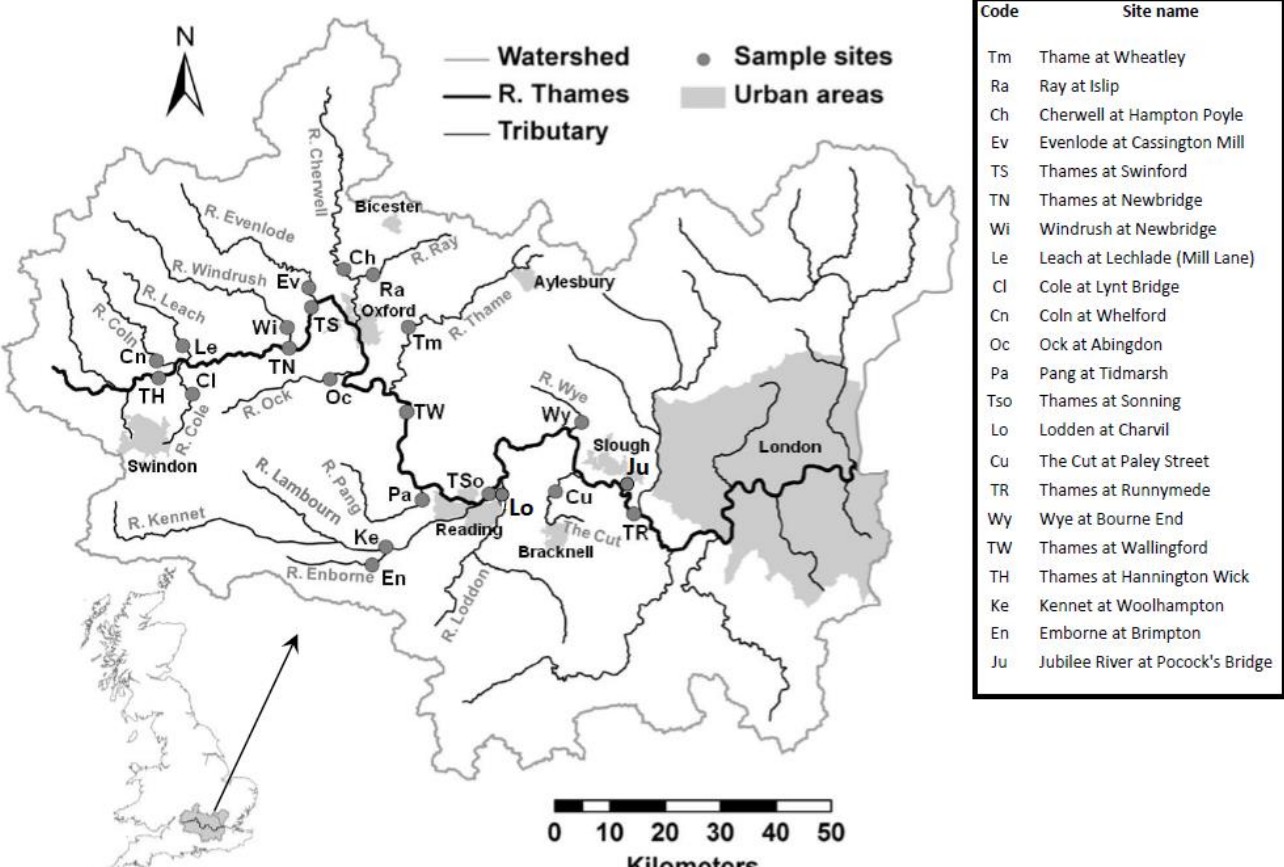

Figure 1





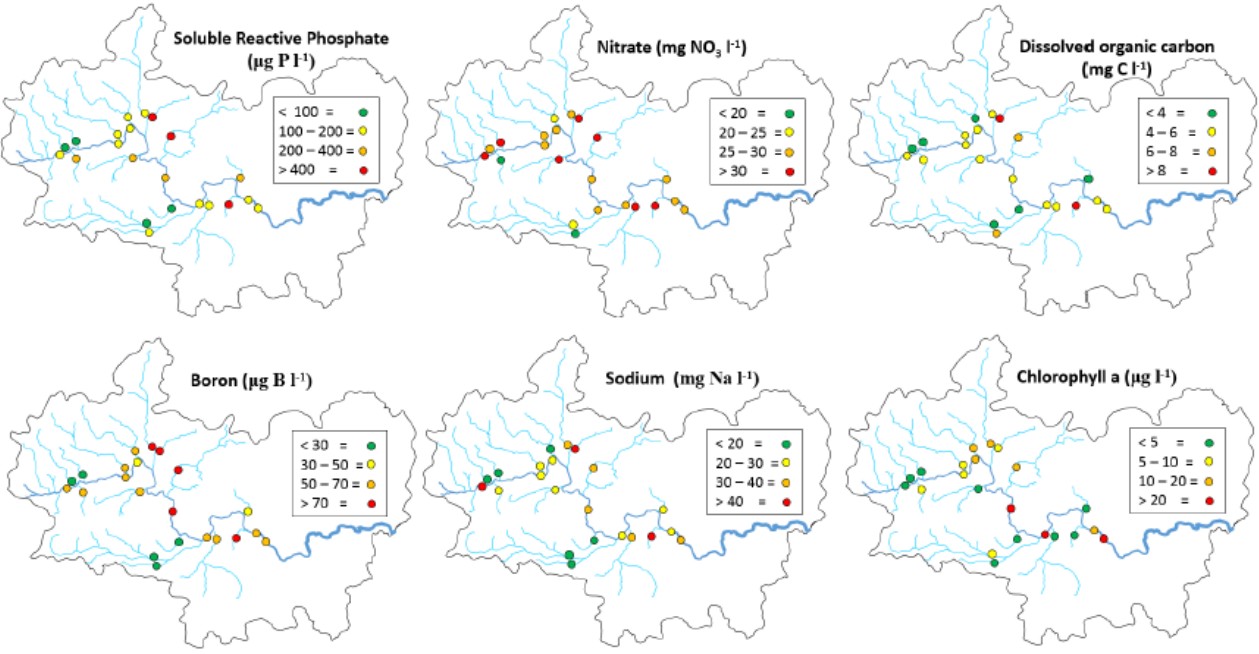

Figure 2



Figure 3



Figure 4





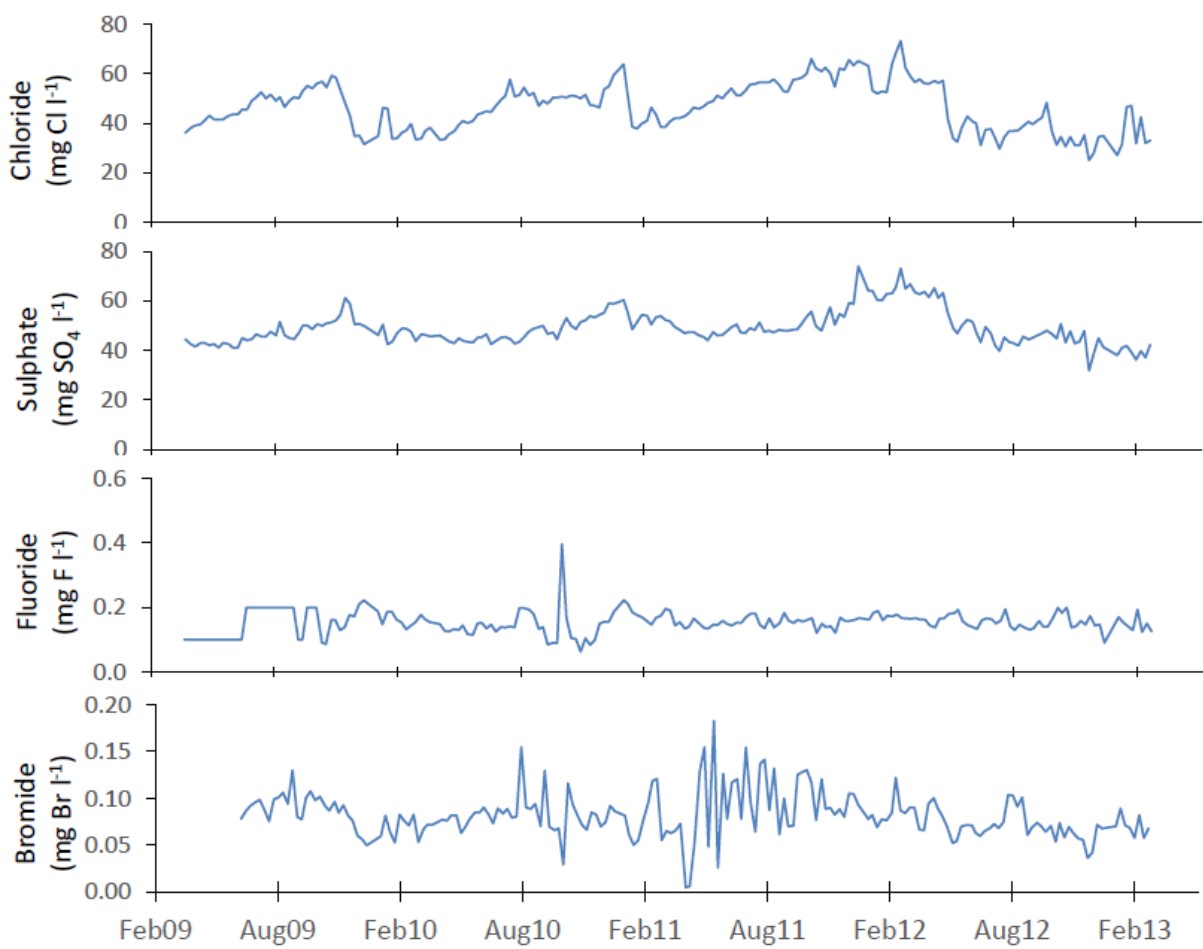

5    Figure 5



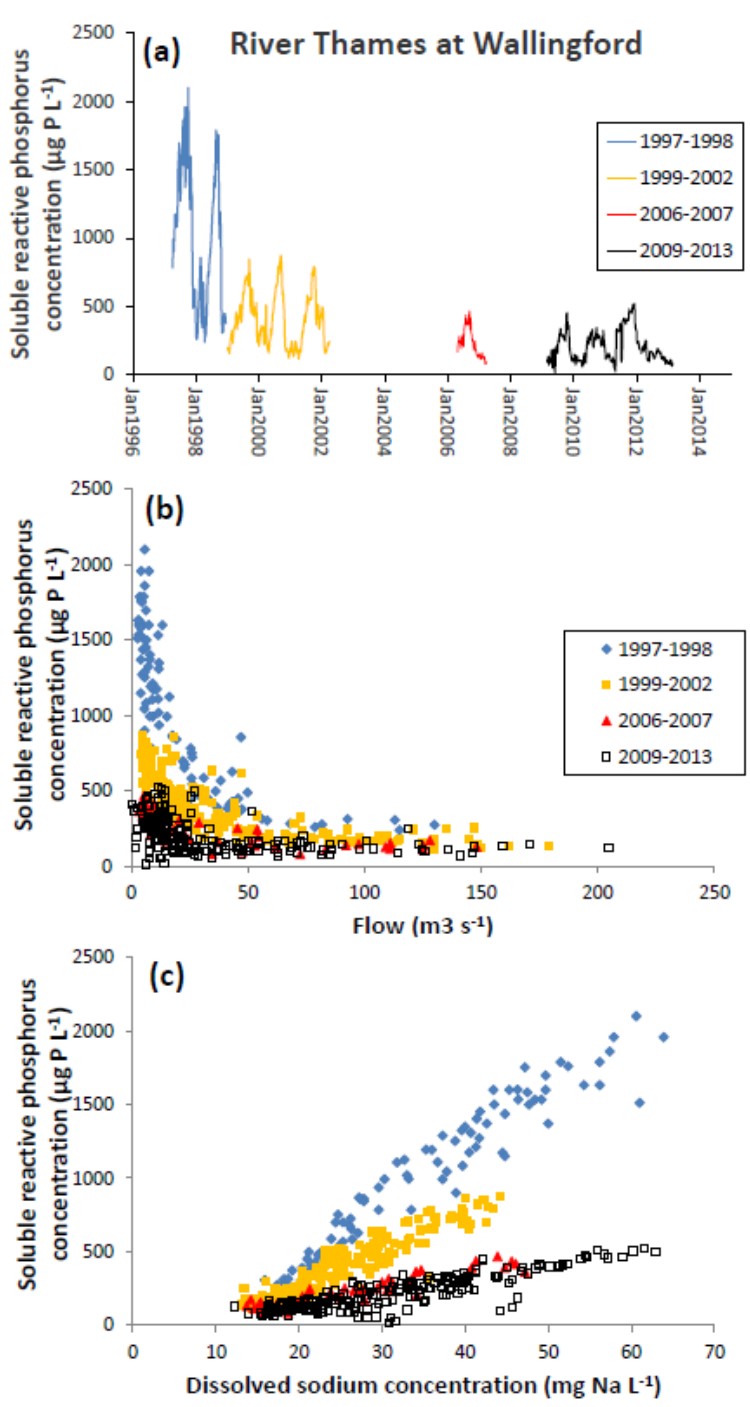

Figure 6.





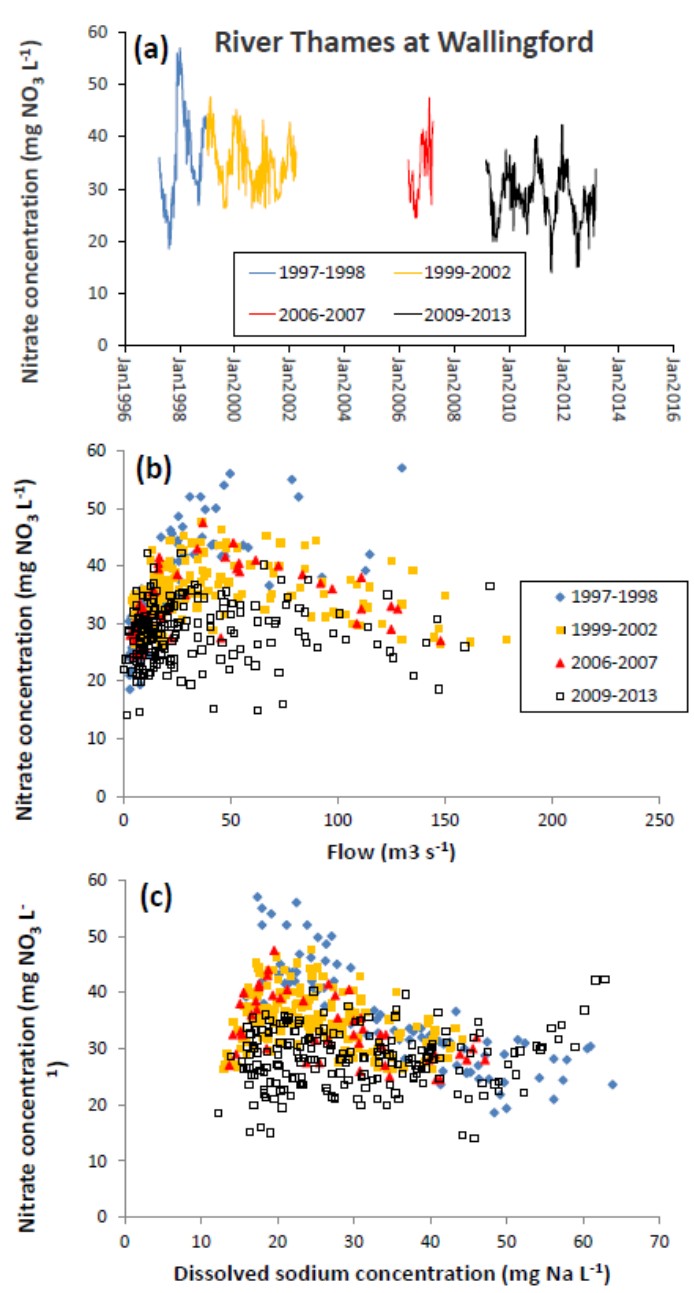

Figure 7.





| Site code in Figure 1 | River name / monitoring location | Monitoring period | Grid reference | Catchment area (km²) | Distance to source (km) | Base flow index | % Land cover | | | | STW PE | STW PE density (PE km⁻²) |
|---|---|---|---|---|---|---|---|---|---|---|---|---|
| | | | | | | | Woodland | Grassland | Arable | Urban / semi-urban | | |
| Cn | River Coln at Whelford | March 2009 - Feb 2013 | SU171991 | 135 | 49 | 0.95 | 15.2 | 38.0 | 42.9 | 2.6 | 5,440 | 40 |
| Cl | River Cole at Lyte Bridge | March 2009 - Feb 2013 | SU210980 | 141 | 33 | 0.54 | 7.2 | 35.9 | 42.0 | 13.1 | 6,620 | 47 |
| Le | River Leach at Lechlade | March 2009 - Feb 2013 | SU228996 | 77 | 34 | 0.79 | 9.7 | 22.8 | 64.2 | 2.0 | 1,540 | 20 |
| Wi | River Windrush at Newbridge | March 2009 - Feb 2013 | SP403014 | 362 | 71 | 0.87 | 13.1 | 34.4 | 45.9 | 4.8 | 46,300 | 128 |
| Ev | River Evenlode at Cassington Mill | March 2009 - Feb 2013 | SP447101 | 427 | 62 | 0.71 | 14.1 | 31.5 | 48.5 | 4.9 | 40,100 | 94 |
| Ch | River Cherwell at Hampton Poyle | March 2009 - Feb 2013 | SP499152 | 566 | 74 | 0.65 | 9.1 | 33.3 | 50.4 | 6.4 | 112,270 | 198 |
| Ra | River Ray at Islip | March 2009 - Feb 2013 | SP527139 | 290 | 33 | 0.57 | 11.3 | 40.1 | 42.6 | 5.3 | 46,020 | 159 |
| Oc | River Ock at Abingdon | March 2009 - Feb 2013 | SU495967 | 255 | 36 | 0.63 | 8.0 | 33.0 | 51.0 | 7.3 | 36,780 | 144 |
| Tm | River Thame at Wheatley | March 2009 - Feb 2013 | SP612050 | 532 | 62 | 0.60 | 9.7 | 32.4 | 35.7 | 8.3 | 153,710 | 289 |
| Pa | River Pang at Tidmarsh | March 2009 - Feb 2013 | SU636747 | 166 | 39 | 0.86 | 17.6 | 27.6 | 46.0 | 4.1 | 4,990 | 30 |
| Ke | River Kennet at Woolhampton | Nov - 2009 to Feb 2013 | SU572667 | 845 | 79 | 0.87 | 13.2 | 30.0 | 48.7 | 4.4 | 96,380 | 114 |
| En | River Enborne at Brimpton | Nov - 2009 to Feb 2013 | SU569649 | 142 | 29 | 0.53 | 24.0 | 38.0 | 28.9 | 6.5 | 11,110 | 78 |
| Lo | River Lodden at Charvil | March 2009 - Feb 2013 | SU779766 | 644 | 53 | 0.80 | 19.6 | 36.7 | 23.1 | 27.3 | 457,890 | 711 |
| Wy | River Wye at Bourne End | March 2009 - Feb 2013 | SU895866 | 134 | 27 | 0.93 | 19.3 | 35.9 | 23.6 | 20.5 | 82,300 | 614 |
| Cu | The Cut at Paley Street | March 2009 - Feb 2013 | SU869762 | 63 | 21 | 0.46 | 20.8 | 32.7 | 9.8 | 35.3 | 103,600 | 1644 |
| TH | River Thames at Hannington | Sept 2009 - Feb 2013 | SU175961 | 566 | 51 | 0.70 | 10.9 | 40.5 | 38.3 | 8.2 | 237,810 | 420 |
| TS | River Thames at Swinford | March 2009 - Feb 2013 | SP442085 | 1627 | 96 | 0.68 | 10.8 | 35.1 | 45.5 | 6.7 | 338,300 | 208 |
| TN | River Thames at Newbridge | March 2009 - Feb 2013 | SP403013 | 1229 | 84 | 0.68 | 10.0 | 35.3 | 45.7 | 7.2 | 290,870 | 237 |
| TW | River Thames at Wallingford | March 2009 - Feb 2013 | SU609902 | 4213 | 147 | 0.65 | 10.3 | 35.6 | 45.1 | 7.3 | 1,027,910 | 244 |
| TSo | River Thames at Sonning | March 2009 - Feb 2013 | SU753758 | 5788 | 181 | 0.68 | 11.5 | 34.9 | 44.3 | 7.5 | 1,586,110 | 274 |
| Ju | Jubilee River at Pocock's Bridge | Jan 2010 to Feb 2013 | SU976783 | 7125 | 231 | 0.72 | 13.2 | 34.0 | 40.6 | 10.3 | 2,613,941 | 367 |
| TR | River Thames at Runnymede | March 2009 - Feb 2013 | TQ006723 | 7170 | 240 | 0.72 | 13.2 | 34.0 | 40.4 | 10.5 | 2,661,370 | 371 |

Table 1.

| Monitoring site | Daily river discharge (m³ s⁻¹) | Water temperature (°C) | pH | Alkalinity (μ equ.l⁻¹) | Suspended solids (mg l⁻¹) | Dissolved fluoride (mg l⁻¹) | Dissolved chloride (mg l⁻¹) | Dissolved bromide (mg l⁻¹) | Dissolved sulphate (mg l⁻¹-SO₄) |
|---|---|---|---|---|---|---|---|---|---|
| River Coln at Whelford | 2.13 | 11.76 | 7.99 | 4248 | 5.4 | 0.13 | 16.7 | 0.04 | 33.7 |
| River Cole at Lynt Bridge | 1.10 | 11.72 | 7.94 | 4342 | 15.0 | 0.19 | 46.5 | 0.08 | 53.4 |
| River Leach at Mill Lane,Lechlade | 0.66 | 11.40 | 7.88 | 4357 | 3.0 | 0.10 | 16.0 | 0.04 | 35.2 |
| River Windrush at Newbridge | 9.75 | 12.17 | 7.97 | 4167 | 10.9 | 0.15 | 40.6 | 0.09 | 53.6 |
| River Evenlode at Cassington Mill | 3.61 | 11.32 | 7.91 | 4027 | 15.5 | 0.12 | 25.7 | 0.05 | 45.7 |
| River Cherwell at Hampton Poyle | 3.70 | 11.47 | 7.91 | 4133 | 13.3 | 0.20 | 54.2 | 0.07 | 65.6 |
| River Ray at Islip | 1.94 | 11.27 | 7.66 | 4096 | 9.6 | 0.18 | 63.3 | 0.11 | 94.6 |
| River Ock at Abingdon | 1.52 | 11.62 | 7.99 | 4704 | 11.0 | 0.20 | 39.2 | 0.09 | 72.0 |
| River Thame at Wheatley | 3.33 | 11.54 | 7.83 | 4468 | 14.0 | 0.22 | 55.4 | 0.08 | 72.0 |
| River Pang at Tidmarsh | 0.53 | 10.82 | 7.90 | 4508 | 8.1 | 0.13 | 24.6 | 0.05 | 19.3 |
| River Kennet at Woolhampton | 8.78 | 11.27 | 7.99 | 4503 | 9.2 | 0.12 | 23.7 | 0.05 | 20.0 |
| River Enborne at Brimpton | 1.27 | 10.46 | 7.75 | 2815 | 9.4 | 0.12 | 34.7 | 0.06 | 26.3 |
| River Lodden at Charvil | 5.39 | 12.15 | 7.82 | 3207 | 7.2 | 0.12 | 60.6 | 0.09 | 47.8 |
| River Wye at Bourne End | 0.84 | 12.30 | 8.07 | 4594 | 13.2 | 0.11 | 42.5 | 0.06 | 20.5 |
| The Cut at Paley Street | 0.38 | 12.52 | 7.58 | 2450 | 9.2 | 0.17 | 94.7 | 0.11 | 99.7 |
| River Thames at Hannington Wick | 4.84 | 11.56 | 7.88 | 3957 | 11.6 | 0.15 | 56.3 | 0.11 | 67.4 |
| River Thames at Newbridge | 9.75 | 12.17 | 7.97 | 4167 | 10.9 | 0.15 | 40.6 | 0.09 | 53.6 |
| River Thames at Swinford | 13.45 | 12.07 | 8.00 | 4070 | 11.5 | 0.14 | 35.4 | 0.08 | 50.2 |
| River Thames at Wallingford | 33.87 | 12.69 | 8.00 | 4149 | 15.3 | 0.17 | 46.2 | 0.08 | 67.5 |
| River Thames at Sonning | 41.33 | 12.31 | 7.96 | 4149 | 11.4 | 0.16 | 40.0 | 0.07 | 50.0 |
| Jubilee River at Pocock's Bridge | 57.66 | 13.12 | 7.95 | 4091 | 8.3 | 0.15 | 44.0 | 0.07 | 47.6 |
| River Thames at Runnymede | 54.11 | 13.01 | 7.93 | 3999 | 11.7 | 0.15 | 46.5 | 0.08 | 48.9 |

5    Table 2.



| Monitoring site | Soluble reactive phosphorus (µg P l⁻¹) | Total dissolved phosphorus (µg P l⁻¹) | Total phosphorus (µg P l⁻¹) | Dissolved nitrate (mg NO₃ l⁻¹) | Dissolved nitrite (mg NO₂ l⁻¹) | Ammonium (mg NH₄ l⁻¹) | Total dissolved nitrogen (mg N l⁻¹) | Dissolved organic carbon (mg l⁻¹) | Dissolved reactive silicon (mg Si l⁻¹) | Chlorophyll-a (µg l⁻¹) |
|---|---|---|---|---|---|---|---|---|---|---|
| River Coln at Whelford | 61 | 68 | 84 | 26.2 | 0.06 | 0.04 | 6.3 | 2.0 | 2.6 | 3.0 |
| River Cole at Lynt Bridge | 228 | 255 | 306 | 18.5 | 0.06 | 0.05 | 4.5 | 5.6 | 6.4 | 5.8 |
| River Leach at Mill Lane,Lechlade | 20 | 24 | 34 | 31.1 | 0.09 | 0.06 | 7.5 | 2.3 | 2.4 | 2.0 |
| River Windrush at Newbridge | 150 | 169 | 206 | 26.7 | 0.06 | 0.05 | 6.5 | 4.4 | 3.3 | 9.6 |
| River Evenlode at Cassington Mill | 175 | 193 | 253 | 24.7 | 0.05 | 0.04 | 6.0 | 3.6 | 2.7 | 12.9 |
| River Cherwell at Hampton Poyle | 124 | 142 | 193 | 25.2 | 0.04 | 0.04 | 6.1 | 5.3 | 3.3 | 14.0 |
| River Ray at Islip | 428 | 472 | 513 | 33.6 | 0.15 | 0.11 | 8.2 | 8.7 | 3.3 | 7.9 |
| River Ock at Abingdon | 253 | 281 | 320 | 30.6 | 0.08 | 0.06 | 7.3 | 5.5 | 7.1 | 3.9 |
| River Thame at Wheatley | 580 | 635 | 712 | 35.1 | 0.13 | 0.24 | 8.6 | 7.5 | 6.5 | 12.4 |
| River Pang at Tidmarsh | 38 | 49 | 68 | 28.1 | 0.05 | 0.04 | 6.7 | 3.1 | 7.0 | 2.8 |
| River Kennet at Woolhampton | 34 | 45 | 78 | 24.1 | 0.05 | 0.05 | 5.9 | 3.2 | 6.8 | 8.2 |
| River Enborne at Brimpton | 115 | 138 | 182 | 17.1 | 0.06 | 0.08 | 4.2 | 6.4 | 6.9 | 2.5 |
| River Lodden at Charvil | 126 | 150 | 211 | 34.4 | 0.09 | 0.08 | 8.4 | 5.8 | 5.5 | 3.9 |
| River Wye at Bourne End | 202 | 234 | 289 | 27.5 | 0.10 | 0.11 | 6.8 | 3.0 | 6.7 | 3.7 |
| The Cut at Paley Street | 506 | 585 | 675 | 83.7 | 0.36 | 0.21 | 21.3 | 9.5 | 6.2 | 4.5 |
| River Thames at Hannington Wick | 181 | 210 | 253 | 32.3 | 0.08 | 0.10 | 7.9 | 5.3 | 3.5 | 3.8 |
| River Thames at Newbridge | 150 | 169 | 206 | 26.7 | 0.06 | 0.05 | 6.5 | 4.4 | 3.3 | 9.6 |
| River Thames at Swinford | 116 | 132 | 171 | 26.0 | 0.05 | 0.05 | 6.3 | 4.0 | 2.9 | 10.8 |
| River Thames at Wallingford | 212 | 241 | 301 | 28.3 | 0.08 | 0.08 | 6.9 | 5.3 | 4.2 | 27.9 |
| River Thames at Sonning | 148 | 172 | 215 | 26.7 | 0.09 | 0.07 | 6.5 | 4.8 | 5.0 | 22.8 |
| Jubilee River at Pocock's Bridge | 133 | 157 | 192 | 26.5 | 0.09 | 0.07 | 6.5 | 5.0 | 5.2 | 18.5 |
| River Thames at Runnymede | 154 | 175 | 222 | 28.1 | 0.10 | 0.09 | 6.9 | 5.1 | 5.1 | 29.5 |

Table 3.

| Monitoring site | Dissolved sodium (mg l⁻¹) | Dissolved potassium (mg l⁻¹) | Dissolved calcium (mg l⁻¹) | Dissolved magnesium (mg l⁻¹) | Dissolved boron (µg l⁻¹) | Dissolved iron (µg l⁻¹) | Dissolved manganese (µg l⁻¹) | Dissolved zinc (µg l⁻¹) | Dissolved copper (µg l⁻¹) |
|---|---|---|---|---|---|---|---|---|---|
| River Coln at Whelford | 9 | 1.7 | 101 | 5.8 | 20 | 8 | 1.4 | 1.9 | 0.5 |
| River Cole at Lynt Bridge | 28 | 5.3 | 110 | 4.4 | 56 | 48 | 10.2 | 3.5 | 1.9 |
| River Leach at Mill Lane,Lechlade | 8 | 1.5 | 109 | 5.1 | 25 | 12 | 2.6 | 1.8 | 0.6 |
| River Windrush at Newbridge | 27 | 5.5 | 104 | 5.2 | 53 | 32 | 6.7 | 3.3 | 1.5 |
| River Evenlode at Cassington Mill | 16 | 3.6 | 102 | 4.2 | 51 | 48 | 6.6 | 2.4 | 1.1 |
| River Cherwell at Hampton Poyle | 36 | 6.2 | 104 | 7.6 | 73 | 57 | 6.8 | 3.3 | 1.5 |
| River Ray at Islip | 49 | 10.5 | 112 | 6.1 | 107 | 121 | 11.9 | 6.8 | 2.5 |
| River Ock at Abingdon | 25 | 5.9 | 127 | 4.6 | 62 | 42 | 7.5 | 3.1 | 1.7 |
| River Thame at Wheatley | 39 | 9.6 | 118 | 5.4 | 87 | 62 | 10.0 | 7.8 | 4.0 |
| River Pang at Tidmarsh | 12 | 2.9 | 108 | 3.2 | 21 | 27 | 2.9 | 3.1 | 1.3 |
| River Kennet at Woolhampton | 13 | 2.4 | 107 | 2.2 | 22 | 19 | 4.3 | 3.0 | 1.0 |
| River Enborne at Brimpton | 18 | 3.6 | 68 | 4.4 | 26 | 142 | 19.6 | 3.6 | 2.2 |
| River Lodden at Charvil | 39 | 7.5 | 83 | 5.3 | 57 | 74 | 17.0 | 5.9 | 2.7 |
| River Wye at Bourne End | 27 | 4.3 | 107 | 1.9 | 35 | 12 | 3.7 | 8.2 | 2.7 |
| The Cut at Paley Street | 71 | 13.6 | 85 | 10.1 | 89 | 91 | 13.3 | 11.0 | 7.5 |
| River Thames at Hannington Wick | 41 | 8.1 | 101 | 5.2 | 65 | 45 | 8.3 | 5.5 | 1.6 |
| River Thames at Newbridge | 27 | 5.5 | 104 | 5.2 | 53 | 32 | 6.7 | 3.3 | 1.5 |
| River Thames at Swinford | 23 | 4.6 | 102 | 5.0 | 47 | 28 | 5.5 | 3.0 | 1.5 |
| River Thames at Wallingford | 30 | 6.5 | 109 | 5.4 | 77 | 42 | 7.1 | 4.3 | 4.0 |
| River Thames at Sonning | 25 | 5.2 | 105 | 4.5 | 58 | 37 | 6.9 | 4.2 | 2.9 |
| Jubilee River at Pocock's Bridge | 28 | 5.4 | 102 | 4.4 | 54 | 29 | 6.1 | 4.1 | 2.8 |
| River Thames at Runnymede | 30 | 5.9 | 101 | 4.6 | 61 | 32 | 5.4 | 4.8 | 2.9 |

Table 4.