# Peer review of "Weekly water quality monitoring data for the River Thames (UK) and its major tributaries (2009-2013): The Thames Initiative research platform"

_Earth System Science Data, 2017_

## Referee Comment (RC1) · Anonymous Referee #1 · 21 May 2018

Review ESSD-2017-139 Thames R water quality

Good data set, well-justified and clearly presented.  Absence of any presentation or discussion of data uncertainties remains a substantial weakness.

Specific comments:

P1, l27: "extensively used for fishing, walking and boating." Technically, users do not walk on the river itself?  Rather, walkers, runners, cyclists, others make extensive use of riverside pathways?  The wider range of users probably adds to the monetary impact alluded to on line 28?  Many of these rivers support abundant, even crowded, fleets of narrow boats?  Alluded to under the generic term 'boating' above?  Their individual or cumulative impact on water quality or river economy (or river morphology?) has a substantial impact?

P3, l5: "… fourth feature of the Thames Initiative has that it needed …" 'was' rather than 'has'?

P4, l2: explain 5-year lag, 2013 to 2018?

P5, l29: explain a STW (sewage treatment works) population estimate.  Total basin sewage load, in terms of BOD over some multi-day period?  Perhaps give the reader / user some perspective on STW PE in the Thames vs other European rivers?

Delay in data access over a weekend, suggests a person involved.  Not the easy one-click access that ESSD users expect.  CEH (or the UK generally) needs to seriously re-think their data access policies and practices.  I have an existing CEH login and it still required Thursday to Monday for data delivery?

Data file in easy-to-use format, clean, well-documented.  I can easily reproduce e.g. Figures 3, 4, 5 etc.

The biggest deficiency arises from the absolute absence of any sense of uncertainty limits or error terms in these data.  Yes, the methods come from Standard Procedure recipes, contained in practical handbooks that many of us keep (rarely used) on our bookshelves.  But each of those methods, from the handbooks or as modified by subsequent cited research, whether for nutrients, cations, chlorophyll, solids, even temperature, has some uncertainty.  Amplify that fundamental uncertainty, often determined among replicates in the laboratory, by any sampling uncertainty peculiar to this location and this particular sampling protocol, across multiple years involving variable weather and changing personnel, and every measurement has some cumulative uncertainty.  The authors of course recognise this but they do not share the information with readers / users.  On page 5 line 16 the authors mention commercial quality control standards but we get no information about which of any or all the samples met those standards, what the authors did if a sample did not pass, etc.  We read earlier (page 5 line 11) about instrument failure (for DOC) and we find occasional gaps in many (most?) data records from most locations.  A figure like Figure 3 should show error bars or an uncertainty band?  Or show us that the error bars or uncertainty bands all consistently fall below some acceptable limits?  River flow data from third party comes with its own uncertainties?  Likewise STW PE has some range of reliability?  For each measured parameter we need at least a plus/minus uncertainty, at 95% CI or 2 std dev.  We also need a paragraph or more on overall uncertainties, including recommendations on what data to avoid when and on how one would improve the overall quality.  What has or will have changed in 2018 compared to 2009?

---

## Referee Comment (RC2) · Anonymous Referee #2 · 1 Jul 2018

General Comments

"Weekly water quality monitoring data for the River Thames (UK) and its major tributaries (2009–2013): The Thames Initiative research platform" is a good introduction to the freely available Thames Initiative data set.

The authors state the objective of the "paper is to present an overview of the . . . data. . . detailing how samples were taken and analysed. . . providing a general description of . . . patterns . . .. and basic interpretation. . ."

[Figure]

They accomplish all aspects of the above objective and provide some interesting interpretations for the patterns in nutrients and chlorophyll-a, data quality and cations. An introduction to a new time series could delve into more in-depth analyses and interpretations. The authors have done this in other works that they cite within the ms. For the purpose of this publication they focus on the general description of patterns and basic interpretation.

With regards to the data itself, the dataset is clean and easy to work with (it was possible to reproduce the time series figures).

I agree with the first reviewer's comment about the need for the authors to identify data uncertainties (error bars, confidence intervals) and the process the authors took for quality checking data.

Specific Comments

Availability of data. I have an account with the Environmental Information Data Centre. Upon my electronic request, an e-mail with a link to download the data was delivered within minutes.

Introduction, page 3 Line 5. "The fourth feature of the Thames Initiative has that it needed to be long term (decadal)..." the manuscript presents results on the years 2009 to 2013 so it seems misleading to present this as a main focus in the introduction.

2.1 Design of Monitoring Programme line 2 page 4. "Data is presented up to the end of Febraury 2013, although the monitoring programme is ongoing." Perhaps the authors should explicitly state that although the monitoring programme is ongoing, freely accessible data is only available until 2013.

2.2 River Sampling lines 5-11. Was there any specific storage of samples between the point of collection and analysis? What temperature were they stored at? Where certain samples kept in the dark?

2.5 Site Characteristics line 28. "The physical characteristics of the catchments were

determined . . . " do these refer to "catchment area" and "distance to source". Might be clearer to rephrase as "Catchment area and distance to source were determined using the Flood Estimation. . ."

2 Sampling and analytical methodology Although no time series for phytoplankton and bacterioplankton is presented in this manuscript, the authors stated "The third feature of the Thames Initiative was that it would characterise aquatic ecology (particularly phytoplankton and bacterioplankton communities) at the same weekly frequency as the water chemistry."

It might be useful if the authors make mention of such a dataset, whether methods for collection of phytoplankton and bacterioplankton are presented elsewhere and whether the data exists elsewhere.

Technical corrections

Abstract Line 15. Suggest changing the word "Comparing" to "combining"

Introduction, page 3 Line 5. Replace the word "has" with "is" in the sentence, "The fourth feature of the Thames Initiative has that it needed to be long term (decadal). . ."

3.2 Spatial data line 11 page 7. "all STWs along its length had tertiary phosphorous stripping installed, . . ." could the authors provide a citation for this

3.3.2 Nutrient data line 28. Suggest adding the word "likely" or "possibly" between the "April 2012," and "due to in-wash"

3.4.1. Phosphorus line 6 page 11. Add bracket before "between 1996"

---

## Author Comment (AC1) · 24 Jul 2018

**Response to Review ESSD-2017-139 Thames R water quality**

The authors would like to thank the reviewer for their comments and suggestions. Please find our responses to these comments in bold below.

Good data set, well-justified and clearly presented. Absence of any presentation or discussion of data uncertainties remains a substantial weakness.

**We are pleased that the reviewer understands the value of this data set. Issues about data uncertainties are addressed below.**

Specific comments:

P1, l27: "extensively used for fishing, walking and boating." Technically, users do not walk on the river itself? Rather, walkers, runners, cyclists, others make extensive use of riverside pathways? The wider range of users probably adds to the monetary impact alluded to on line 28? Many of these rivers support abundant, even crowded, fleets of narrow boats? Alluded to under the generic term 'boating' above? Their individual or cumulative impact on water quality or river economy (or river morphology?) has a substantial impact?

**The text has been altered as suggested, to give further details of the activities that take place within the river and along its riverside paths.**

P3, l5: "… fourth feature of the Thames Initiative has that it needed …" 'was' rather than 'has'?

**This typo has been corrected.**

P4, l2: explain 5-year lag, 2013 to 2018?

**Added to manuscript. Further data will be periodically released through the same data portal, and so this paper will remain relevant.**

P5, l29: explain a STW (sewage treatment works) population estimate. Total basin sewage load, in terms of BOD over some multi-day period? Perhaps give the reader / user some perspective on STW PE in the Thames vs other European rivers?

**The definition of STW-PE has been added. The paper already provides PE values for each individual study site in Table 1. The values for Thames at Runnymede (2.66 million) provides the reader with the sewered population within the Thames study catchment. We believe that providing contrasts with other international catchments is beyond the scope of the paper. If we provide comparisons of PEs and population densities, we would also need to contrast the other catchment statistics and water quality data, which would make the paper unwieldy.**

Delay in data access over a weekend, suggests a person involved. Not the easy one-click access that ESSD users expect. CEH (or the UK generally) needs to seriously re-think their data access policies and practices. I have an existing CEH login and it still required Thursday to Monday for data delivery? Data file in easy-to-use format, clean, well-documented. I can easily reproduce e.g. Figures 3, 4, 5 etc.

**The system is automatic, without staff intervention. I assume that the problem may have been due to server maintenance over the weekend at the time of downloading, and the Reviewer was just unlucky. I have retried the link and received the data within 2 minutes. Reviewer 2 also had a good experience of accessing the data.**

The biggest deficiency arises from the absolute absence of any sense of uncertainty limits or error terms in these data. Yes, the methods come from Standard Procedure recipes, contained in practical handbooks that many of us keep (rarely used) on our bookshelves. But each of those methods, from the handbooks or as modified by subsequent cited research, whether for nutrients, cations, chlorophyll, solids, even temperature, has some uncertainty. Amplify that fundamental uncertainty, often determined among replicates in the laboratory, by any sampling uncertainty peculiar to this location and this particular sampling protocol, across multiple years involving variable weather and changing personnel, and every measurement has some cumulative uncertainty. The authors of course recognise this but they do not share the information with readers / users.

**We accept that this is an omission, and now include information on Limits of Quantification and data uncertainty in Table S1.**

On page 5 line 16 the authors mention commercial quality control standards but we get no information about which of any or all the samples met those standards, what the authors did if a sample did not pass, etc.

**Text has been added to the manuscript in a new Quality Control section (2.4) to fully describe the external and internal QC procedure, and data uncertainty.**

We read earlier (page 5 line 11) about instrument failure (for DOC) and we find occasional gaps in many (most?) data records from most locations.

**The gap in data for DOC is due to the instrument failure, as set out in the text. Apart from this, the data set is near continuous.**

A figure like Figure 3 should show error bars or an uncertainty band? Or show us that the error bars or uncertainty bands all consistently fall below some acceptable limits?

**We have now added Table S1 to show limits of quantification and uncertainty for all analytical methods, in terms of 2 x standard deviations. We disagree that the graphs should have error bars, as this would imply that we had been taking replicate samples from each site each week, which is not the case, and also rarely (if ever) done in this type of river water quality study. This is because rivers of this type are considered to be very well mixed.**

River flow data from third party comes with its own uncertainties? Likewise STW PE has some range of reliability? For each measured parameter we need at least a plus/minus uncertainty, at 95% CI or 2 std dev. We also need a paragraph or more on overall uncertainties, including recommendations on what data to avoid when and on how one would improve the overall quality. What has or will have changed in 2018 compared to 2009?

**We consider that describing the errors in these third-party data sets is beyond the scope of this paper.  The UK hydrometric network is highly regarded for data quality.  The text makes it clear that the STW Population Estimates are (by definition an estimate, to give the reader an indication of the population and sewage loading pressures.  In terms of what has changed over the last 10 years of monitoring, this will be the focus of a paper to be published in the coming year, hopefully alongside the next batch of water quality data.**

---

## Author Comment (AC2) · 24 Jul 2018

The authors would like to thank the reviewer for their comments and suggestions. Please find our responses to these comments in bold below.

General Comments

"Weekly water quality monitoring data for the River Thames (UK) and its major tributaries (2009–2013): The Thames Initiative research platform" is a good introduction to the freely available Thames Initiative data set. The authors state the objective of the "paper is to present an overview of the data detailing how samples were taken and analysed, providing a general description of patterns and basic interpretation"

They accomplish all aspects of the above objective and provide some interesting interpretations for the patterns in nutrients and chlorophyll-a, data quality and cations. An introduction to a new time series could delve into more in-depth analyses and interpretations. The authors have done this in other works that they cite within the ms. For the purpose of this publication they focus on the general description of patterns and basic interpretation.

With regards to the data itself, the dataset is clean and easy to work with (it was possible to reproduce the time series figures).

**We would like to thanks the reviewer for their positive comments about the manuscript and data set.**

I agree with the first reviewer's comment about the need for the authors to identify data uncertainties (error bars, confidence intervals) and the process the authors took for quality checking data.

**I have now added a table as supplementary data (S1), giving information on Limits of Quantification and uncertainty. A section on quality control procedures has been added to the manuscript (section 2.4).**

Specific Comments

Availability of data. I have an account with the Environmental Information Data Centre. Upon my electronic request, an e-mail with a link to download the data was delivered within minutes.

**Really pleased that the data download works seamlessly!**

Introduction, page 3 Line 5. "The fourth feature of the Thames Initiative has that it needed to be long term (decadal)" the manuscript presents results on the years 2009 to 2013 so it seems misleading to present this as a main focus in the introduction.

2.1 Design of Monitoring Programme line 2 page 4. "Data is presented up to the end of February 2013, although the monitoring programme is ongoing." Perhaps the authors should explicitly state that although the monitoring programme is ongoing, freely accessible data is only available until 2013.

**The text has been altered (Page 3, line 8) to make it clear that the aim is for the Thames Initiative to DEVELOP into a decadal study. It is also made clear that the data set is 2009-2013, and that the next batch of data will be released in the near future through the same data portal (Page 4, line 7-8).**

2.2 River Sampling lines 5-11. Was there any specific storage of samples between the point of collection and analysis? What temperature were they stored at? Where certain samples kept in the dark?

**This has been added to page 4, line 18.**

2.5 Site Characteristics line 28. "The physical characteristics of the catchments were determined" Do these refer to "catchment area" and "distance to source". Might be clearer to rephrase as "Catchment area and distance to source were determined using the Flood Estimation"

**Change made.**

2 Sampling and analytical methodology Although no time series for phytoplankton and bacterioplankton is presented in this manuscript, the authors stated "The third feature of the Thames Initiative was that it would characterise aquatic ecology (particularly phytoplankton and bacterioplankton communities) at the same weekly frequency as the water chemistry." It might be useful if the authors make mention of such a dataset, whether methods for collection of phytoplankton and bacterioplankton are presented elsewhere and whether the data exists elsewhere.

**The method used to carry out the weekly algal and bacterial analysis (flow cytometry) has been added to the text along with a reference to the methods paper.**

Technical corrections
Abstract Line 15. Suggest changing the word "Comparing" to "combining"

**Change made**

Introduction, page 3 Line 5. Replace the word "has" with "is" in the sentence, "The fourth feature of the Thames Initiative has that it needed to be long term (decadal)"

**Change made**

3.2 Spatial data line 11 page 7. "all STWs along its length had tertiary phosphorous stripping installed" could the authors provide a citation for this

**Citation provided**

3.3.2 Nutrient data line 28. Suggest adding the word "likely" or "possibly" between the "April 2012," and "due to in-wash"

**Word added.**

3.4.1. Phosphorus line 6 page 11. Add bracket before "between 1996"

**Done.**